# Aptardi predicts polyadenylation sites in sample-specific transcriptomes using high-throughput RNA sequencing and DNA sequence

Ryan Lusk [1✉], Evan Stene [2], Farnoush Banaei-Kashani[2], Boris Tabakoff[1], Katerina Kechris[3] & Laura M. Saba [1]

Annotation of polyadenylation sites from short-read RNA sequencing alone is a challenging computational task. Other algorithms rooted in DNA sequence predict potential poly-adenylation sites; however, in vivo expression of a particular site varies based on a myriad of conditions. Here, we introduce aptardi (alternative polyadenylation transcriptome analysis from RNA-Seq data and DNA sequence information), which leverages both DNA sequence and RNA sequencing in a machine learning paradigm to predict expressed polyadenylation sites. Specifically, as input aptardi takes DNA nucleotide sequence, genome-aligned RNA-Seq data, and an initial transcriptome. The program evaluates these initial transcripts to identify expressed polyadenylation sites in the biological sample and refines transcript 3'-ends accordingly. The average precision of the aptardi model is twice that of a standard tran-scriptome assembler. In particular, the recall of the aptardi model (the proportion of true polyadenylation sites detected by the algorithm) is improved by over three-fold. Also, the model—trained using the Human Brain Reference RNA commercial standard—performs well when applied to RNA-sequencing samples from different tissues and different mammalian species. Finally, aptardi's input is simple to compile and its output is easily amenable to downstream analyses such as quantitation and differential expression.

[1] Department of Pharmaceutical Sciences, University of Colorado Anschutz Medical Campus, Aurora, CO, USA. [2] Department of Computer Science and Engineering, University of Colorado Denver, Denver, CO, USA. [3] Department of Biostatistics and Informatics, University of Colorado Anschutz Medical Campus, Aurora, CO, USA. ✉email: ryan.lusk@cuanschutz.edu

Alternative polyadenylation (APA) is gene regulation mechanism by which a single gene encodes multiple RNA isoforms with different polyadenylation (polyA) sites[1] (i.e., different transcription stop sites/3′ termini). Most APA sites lead to identical protein products but variable 3′ untranslated region lengths[2]. APA has been associated with disease through many transcripts displaying APA (e.g., cardiac hypertrophy[3], oculopharyngeal muscular dystrophy[4,5], breast cancer, and lung cancer[6]) and APA in an individual transcript (e.g., Fabry disease[7], amyotrophic lateral sclerosis[8], metachromatic leukodystrophy[9], and facioscapulohumeral muscular dystrophy[10]). Furthermore, differences in expression of APA transcripts have been implicated in diseases[11] and are recognized as risk factors in complex diseases[12]. Indeed, research suggests individual susceptibility to complex diseases is mainly owing to variation in gene regulation processes—such as APA—rather than variation in protein-coding sequence[13–16]. APA's impact is expected given that it is pervasive, with >70% of human genes subjected to APA[17], and also far-reaching, as it modulates mRNA stability, translation, nuclear export, and cellular localization, as well as the localization of the encoded protein[2,18]—often times through differences in microRNA binding availability.

APA patterns are tissue specific[19,20], and "choice" of polyA sites can be influenced by physiological, environment, and disease states[1,21]. This dynamic may explain—at least in part—why polyA sites are often under annotated[22] and, furthermore, why (the often times sparse) prior annotation is typically not relevant to the given set of experimental conditions[23]. As a result, polyA sites often need to be re-defined for the sample(s) of interest to gain insight into the role of APA in various processes and diseases (e.g., are certain APA transcripts biomarkers of, or therapeutic targets for, a given disease state?). There are three broad sequencing technologies utilized to identify polyA sites: (1) short-read RNA sequencing (RNA-Seq), (2) direct 3′-end RNA sequencing, and (3) DNA sequence, but each possesses inherent limitations for sample-specific identification of polyA sites.

Next-generation RNA-Seq has become the standard technology to profile the expressed transcriptome. The resulting short reads are used by transcriptome assemblers to produce a genome-scale, sample-specific transcriptome map. Transcriptome assembly has proven a powerful approach to assess the transcriptome, but accurate determination of polyA sites from short-read RNA-Seq alone is a known shortcoming[1,24–27]. Unlike splice junctions that can be precisely located via reads that span the junctions, polyA sites are characterized by a gradual drop off in coverage[28]. For assemblers that harness prior annotation to guide the reconstruction, often the annotated polyA site assumed by the assembler is not correct[22]. Although many transcriptome assemblers have been developed—each with its own design—to our knowledge none have demonstrated competence at annotating 3′-ends. Some assemblers, e.g., Cufflinks[29]/StringTie[22], construct a minimum path RNA-Seq cover to the position where there is zero read coverage to annotate the 3′-end of a transcript[28,30]; but, since reads can be derived from precursor mRNA[31], this often results in an overestimation of polyA sites. Others, e.g., Scripture[32], calculate scan statistics above genomic background to define transcript structures, but this approach tends to produce biased estimates of polyA sites and, in general, is not well-suited for defining 3′-ends[33]. Importantly, these strategies are only capable of producing a single transcript stop site per intron chain structure, which tends to be the distal polyA site, thereby missing embedded proximal polyA sites, i.e., APA isoforms[28]. The challenge of accurately identifying polyA sites is apparent to both the developers and those evaluating assemblers by way of allowing for error at 3′-end predictions when assessing accuracy[22,34].

Acknowledging the challenges of annotating polyA sites and design shortcomings of transcriptome assemblers to do so, researchers have developed supplemental tools to characterize APA dynamics from RNA-Seq. Chen et al.[35] provided a comprehensive critical review of these methods which we will briefly highlight here. There are three main methods for characterizing polyA sites and/or quantifying APA dynamics. Those that require a priori annotated polyA sites, e.g., MISO[36], QAPA[37], and PARQ[38], cannot identify de novo polyA sites. Others such as Kleat[39] and ContextMap 2[40] utilize reads with strings of adenosines not derived from a DNA template, i.e., polyA tails. However, studies have demonstrated that polyA reads are scarce in RNA-Seq data[41,42], resulting in low sensitivity and missing more weakly expressed polyA sites. Finally, those that consider fluctuations in read coverage near the 3′-ends of transcripts, e.g., DaPars[43], APAtrap[44], and TAPAS[45], are largely interested in single gene APA switching and/or quantifying differential APA usage between two groups of samples rather than producing a complete transcriptome. Also—as noted by Chen et al.[35]—these tools are not user-friendly; specific input formats are required and outputs are not readily integrable into downstream studies.

An alternative approach is to directly capture 3′-ends of mRNA with sequencing technology e.g., PolyA-Seq[46], 3′ READS[47], PAS-Seq[27], etc. (see Shi[17], Elkon et al.[48], and Ji et al.[49] for a complete review). These methods are accurate at characterizing the genomic locations of polyA sites; however, whereas RNA-Seq data are widely available, 3′-sequencing data represents only a small fraction of available sequencing data and is costly and labor intensive to produce[28,35].

A final category of algorithms has sought to capitalize on the wealth of research connecting specific strings of DNA nucleotides, or DNA sequence elements, to polyadenylation (see Tian and Gaber[50] for a detailed review). Most of these methods, e.g., DeepPASTA[51], Omni-PolyA[52], and Conv-Net[53], deploy machine learning but conspicuously do not consider in vivo expression.

To overcome current limitations, we introduce aptardi (APA transcriptome analysis from RNA-Seq data and DNA sequence information). Aptardi leverages the information afforded by DNA nucleotide sequence information (from the appropriate reference genome) and RNA-Seq, as well as the predilection of transcriptome assemblers to accurately characterize splice junctions, in a use-all-data, multi-omics approach to create a modified, sample-specific transcriptome that includes information on expressed polyA sites (Fig. 1). Specifically, harnessing the power of (supervised) machine learning, we trained aptardi to detect polyA sites from DNA nucleotide sequence and RNA-Seq read coverage by training on polyA sites identified by 3′ sequencing. Using what it learned, aptardi makes predictions from DNA sequence and RNA-Seq alone, alleviating the burden of generating 3′-sequencing data. The program evaluates initial transcripts in the input original transcriptome to identify expressed polyA sites in the biological sample and refines transcript 3′-ends accordingly and outputs its results to a modified transcriptome (as a General Feature Format [GTF] file). Additionally, aptardi's input is simple to compile and its output is easily amenable to downstream analyses such as quantitation and differential expression.

## Results

**Construction of multi-omics model for identification of polyadenylation sites.** The initial data set used for developing the aptardi model was derived from Human Brain Reference[54] (HBR) RNA using Illumina's TruSeq stranded mRNA sample preparation kit to generate 100 base, paired-end reads[55]. The transcriptome reconstruction contained 113,923 transcripts (excluding those from scaffold chromosomes) with 94,369 unique transcript termini, and the corresponding PolyA-Seq data contained 94,322 polyA sites in this sample. Throughout this manuscript, we refer

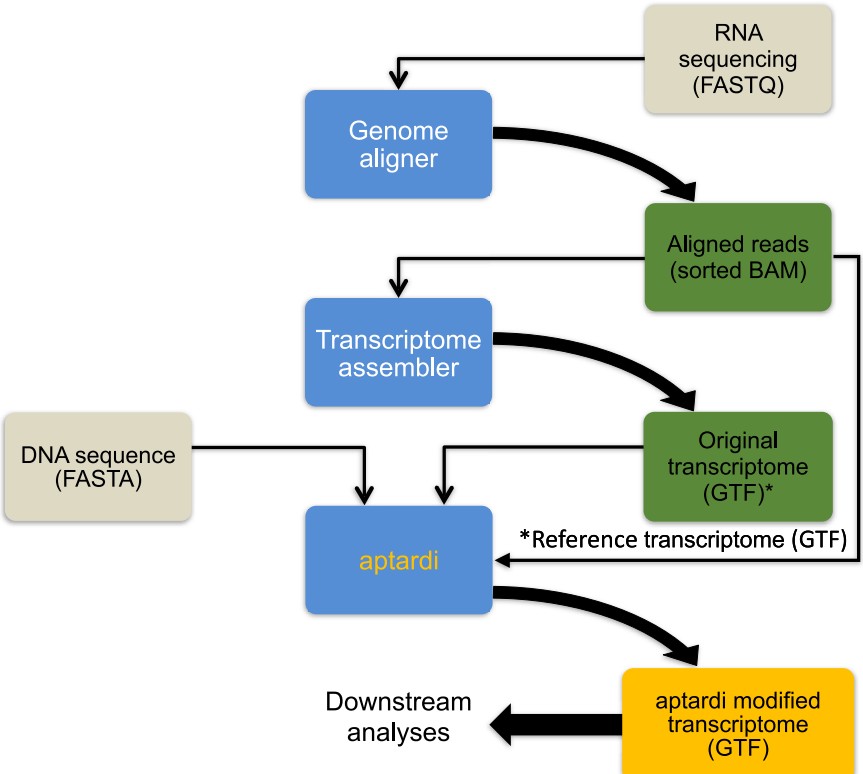

**Fig. 1 Overview for using aptardi.** Aptardi requires three files as input: (1) FASTA file of DNA sequence with headers by chromosome, (2) sorted Binary Alignment Map (BAM) file of reads aligned to the genome, and (3) General Feature Format (GTF) file of transcript structures. Blue boxes represent software. Yellow writing/boxes indicate aptardi incorporation. Note transcript structures can be derived from a reference transcriptome (i.e., Ensembl annotation) in lieu of the original transcriptome generated from a transcriptome assembler.

to the polyA sites identified from PolyA-Seq[46] as "true" polyA sites to distinguish them from polyA sites predicted by a computational algorithm, but we acknowledge that there are false negatives and false positives among the PolyA-Seq derived polyA sites. After transcript processing (see Methods) and integration with the PolyA-Seq data—generated from the same HBR RNA—70,748 transcript models with 0–50 true polyA sites per transcript model were used for learning and evaluating the aptardi prediction model. The modified 3′-terminal exons of these transcript models were binned into 100 base increments for machine learning, and 14 RNA-Seq features, 12 DNA sequence-related features, and one feature derived from the original transcriptome were calculated for each bin.

As examples of DNA sequence-derived features, the presence of the three strong polyA signals (5′-AATAAA-3′, 5′-ATTAAA-3′, and 5′-AGTAAA-3′) in each 100 base bin as a function of whether the bin also contained a polyA site are shown in Fig. 2a. Each strong polyA signal demonstrated enrichment (AATAAA: $\chi^2 = 80,837$, $p$ value $< 0.0001$; ATTAAA: $\chi^2 = 15,012$, $p$ value $< 0.0001$; AGTAAA: $\chi^2 = 1378$, $p$ value $< 0.0001$). For instance, of the 100 base bins that possessed a true polyA site via PolyA-Seq, over half also possessed AATAAA. In contrast, only ~10% of bins that did not contain a polyA site had the AATAAA signal. This enrichment was observed for all binary features, i.e., all the DNA sequence features and the original transcriptome end location feature (Supplementary Fig. 1a). The strong polyA signals were also independently associated with the presence of a polyA site. Likewise, the distribution of the quantitative RNA-Seq features, e.g., the inter-bin RNA-Seq features (Fig. 2b) differed based on the presence or absence of a true polyA site (although no one feature distinguishes the true polyA sites perfectly) and this was also seen for the intra-bin RNA-Seq features (Supplementary Fig. 1b). Furthermore, features derived from DNA

sequence and RNA-Seq were independent of one another across omics type but were often correlated within an omics category (Fig. 2c).

When aptardi was built using only features derived from RNA-Seq or only features derived from DNA sequence, the average precision (AP) in the testing data set was significantly greater than simply relying on the polyA sites identified in the original transcriptome (Fig. 2d). Furthermore, when the RNA-Seq features and the DNA sequence features were combined, the multi-omics model had higher AP than either single-omics model (multi-omics AP = 0.58, DNA-only AP = 0.41, RNA-only AP = 0.44). Using a specific prediction threshold (probability $> 0.5$), the precision in the multi-omics model (0.74) increased from the DNA-only model (0.65) but only modestly increased from the RNA-only model (0.71); however, the recall dramatically improved compared to both single-omics models (multi-omics recall = 0.39, DNA-only recall = 0.18, RNA-only = 0.24; Fig. 2e). The F-measure was similarly greater in the multi-omics model than either single-omics model (multi-omics = 0.51, DNA-only = 0.28, RNA-only = 0.36). In addition, performance results were consistent across five random splits of the data for training/validation/testing, and the results displayed above are the averages across the five splits.

The relative contributions of the individual DNA sequence features were further explored by generating aptardi models for each DNA sequence feature that either (1) included all other features but the DNA sequence feature or (2) removed all other DNA-derived features but the DNA sequence feature (i.e., all of the RNA-Seq features and the original transcriptome feature were also still included) and evaluating performance on the testing split. Unsurprisingly, the greatest reduction in performance from leaving out any single DNA sequence feature came from removing the canonical polyA signal (AP = 0.52 vs 0.58 in full model, F-measure = 0.45 vs

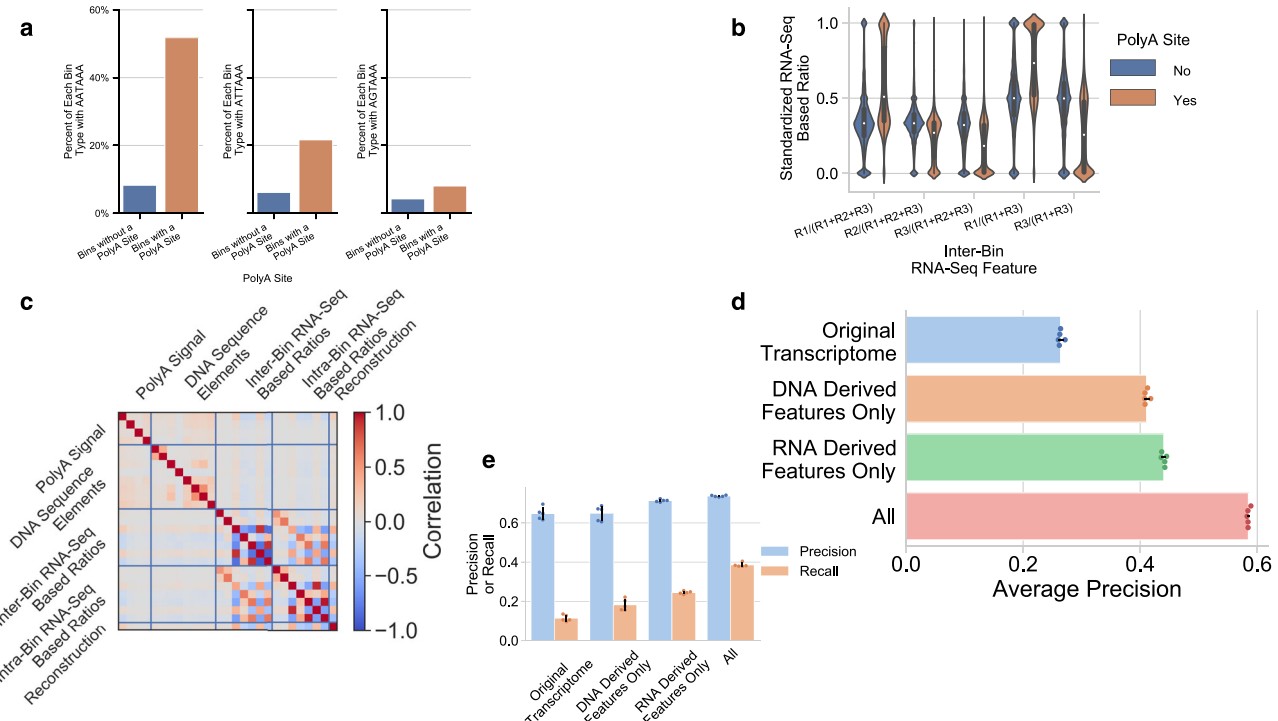

**Fig. 2 DNA sequence and RNA-sequencing (RNA-Seq) features are individually associated with polyadenylation (polyA) sites. a** The percent of 100 base bins containing each of the three strong polyA signals stratified by the bin not containing (blue) or containing (orange) a polyA site. **b** Distribution of the inter-bin RNA-Seq features for each 100 base bin stratified by the bin not containing (blue) or containing (orange) a polyA site (RNA-Seq ratio features were standardized using the training set). **c** RNA-Seq features and DNA sequence features display little correlation (two-sided Pearson Product-Moment) across omics type. The combination of RNA-Seq information and DNA sequence information improves **d** average precision, and **e**, precision and recall at a specific prediction threshold (probability >0.50) over each separately. For both **d** and **e**, data are presented as mean values ±standard deviation on the test set (n = 5 random train-validate-test splits). Data shown are from the Human Brain Reference data set.

0.52 in full model), and the greatest improvement by including any single DNA sequence feature was from this feature as well (AP = 0.54 vs 0.47 without any DNA-derived features, F-measure = 0.47 vs 0.39 without any DNA-derived features).

**Evaluation of the generalizability of aptardi**. To evaluate the generalizability of the aptardi prediction model, we asked two questions: (1) does the performance of the aptardi prediction model, built on the HBR data set, remain consistent across diverse data sets, and (2) are the performances of prediction models built on alternative data sets comparable to the aptardi prediction model (built from the HBR data set)? To answer these questions, we analyzed four alternative data sets. These data sets were chosen because they had sufficient similarities and differences to assess the applicability of the aptardi prediction model (Supplementary Table 1). Namely, an additional Human Brain Reference RNA data set was included that was derived from the same Human Brain Reference RNA sample but processed and sequenced in another laboratory (2nd HBR), and this laboratory also produced another data set we included from Universal Human Reference (UHR) RNA[56]. To include a cross-species comparison and to examine similar tissue across two genetically different individuals, we also used data derived from two inbred rat strains; the congenic Brown Norway strain with polydactyly-luxate syndrome (BNLx/Cub; BNLx) and the spontaneously hypertensive rat strain (SHR/OlaIpcv; SHR). All true polyA sites were derived from 3′ sequencing PolyA-Seq data; true polyA sites for the HBR and UHR data sets were from the same corresponding RNA, whereas the true polyA sites for the two rat data sets were derived from Sprague Dawley rat brain RNA[46].

We first examined whether users can confidently apply the aptardi prediction model, built from the HBR data set, on their own data sets (i.e., on a data set not used to train the model) by comparing its performance on the four alternative data sets not used to train the model. The AP of the HBR-based aptardi prediction model across the four other data sets ranged from 0.55 to 0.63, whereas the AP of this HBR aptardi prediction model on its own HBR data set was 0.65 (Fig. 3; orange bars). Specifically, its performance on the other human RNA samples (2nd HBR and UHR) only differed in AP by two percentage points (AP = 0.63 for each), but on the BNLx and SHR rat brain data sets the HBR-based aptardi prediction model performed more modestly (AP = 0.55 for each). Similar results were observed for F-measures (HBR = 0.56, 2nd HBR = 0.51, UHR = 0.53, BNLx = 0.47, SHR = 0.48). The major differences between the two rat data sets and the HBR data set include species, strandedness of the library preparation (rat samples were unstranded), and the inexact matching between RNA-Seq and PolyA-Seq RNA sources.

Also for the four alternative data sets, we built data set-specific prediction models and compared their performance on their own data set to the performance of the HBR-based aptardi prediction model on the given data set to demonstrate the robustness of the machine learning pipeline used to build the aptardi prediction model. For all four data sets, the increase in AP when the same data set for training the prediction model is used for evaluating the prediction model (as opposed to the performance of the HBR aptardi prediction model on the same given data set) was minimal, i.e., ≤2 percentage points (Fig. 3). Furthermore, the similarity (within two percentage points) of the AP between the training, testing, and analysis (i.e., not merging transcripts; see Methods) sets demonstrate the aptardi prediction model is not prone to overfitting

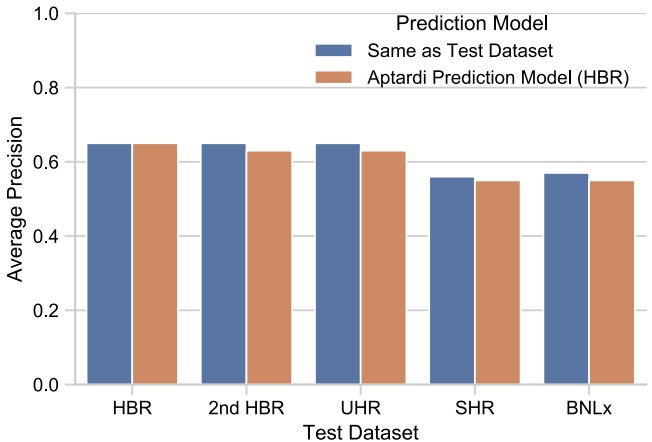

**Fig. 3 The machine learning pipeline used to build aptardi is robust to different data sets and the aptardi prediction model generated from the Human Brain Reference data set is applicable across diverse data sets.** Blue bars indicate the performance of the data set-specific prediction model on its own data set, i.e., the model was built and evaluated on a single data set. Orange bars represent the performance of the aptardi prediction model—built from the Human Brain Reference data set—on the given data set (x axis).

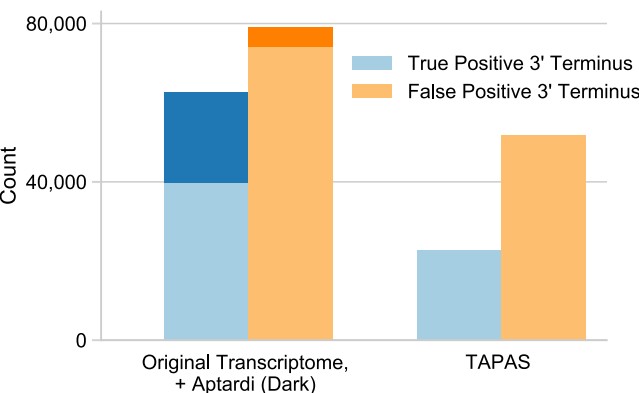

**Fig. 4 Incorporating aptardi transcripts into the original transcriptome improves the ratio of true positive to false positive 3′ termini compared with the original transcriptome and compared with the Tool for Alternative Polyadenylation site AnalysiS (TAPAS) analysis on the original transcriptome.** Results from transcripts added by aptardi to the original transcriptome are shaded in dark. Transcripts whose 3′ terminus was plus or minus 100 bases of a true polyadenylation site from PolyA-Seq data were considered a true positive and otherwise counted as a false positive. Data shown are from the Human Brain Reference data set.

and, when these intra- and inter- model/data set comparisons with the HBR data set were extended to each of the four data sets, similar results were achieved (Supplementary Fig. 2).

**Performance of aptardi on other true polyadenylation sites data sets.** Other polyA site data sets in the literature report polyA sites aggregated from multiple samples and sources, such as PolyASite 2.0[57] and PolyA_DB 3[58]. Although aptardi was designed to make sample-specific predictions, we evaluated the performance of the aptardi prediction model (built from HBR) on these more extensive true polyA sites data sets using the HBR RNA-Seq data and hg38/GRCh38 human genome. For each polyA site cluster listed by PolyASite 2.0, the representative polyA site was used. PolyASite 2.0 and PolyA_DB 3 contained 569,005 and 289,998 annotated polyA sites leading to 102,774 and 86,685 polyA site bins, respectively (compared with 42,977 polyA site bins using the HBR PolyA-Seq data) out of the 778,166 bins produced from the transcript processing of HBR. As expected, the precision increased (PolyASite 2.0 = 0.85, PolyA_DB_3 = 0.88) and the recall decreased (PolyASite 2.0 = 0.18, PolyA_DB = 0.22) compared with the aptardi prediction model's performance on its testing data set (precision = 0.71, recall = 0.41).

**Improvement of 3′-end annotation in the transcriptome map by aptardi.** The overarching goal of aptardi is to yield an updated, sample/experiment-specific transcriptome map from the original transcriptome with more accurately annotated 3′-ends of expressed polyadenylated transcripts. As such, aptardi was primarily benchmarked by comparing how it improved upon the reconstruction generated by the popular assembler StringTie[22]. The StringTie assembly also incorporated Ensembl[59] (v.99) annotation, which helps guide its reconstruction—especially at 3′-ends. Note that aptardi outputs all original transcript structures from the input transcriptome (i.e., original transcriptome) in addition to those annotations identified by the program.

Of the 113,923 transcripts in the original transcriptome from the HBR sample (i.e., StringTie used the HBR sample RNA-Seq data and existing Ensembl annotation to generate the original transcriptome), only 39,842 (35%) had a 3′ terminus that

corresponded to a true polyA site (±100 bases). When the aptardi prediction model was incorporated, 27,853 transcript annotations were added to the original transcriptome where the polyA site/3′ terminus differed from its original transcript structure. Of these additional 27,853 transcripts, 22,846 (82%) matched the location of a true polyA site in the HBR PolyA-Seq data (±100 bases), meaning the majority of aptardi transcript structures incorporated into the original transcriptome had accurate polyA site annotation (Fig. 4). Furthermore, the confusion matrix of predictions made by the aptardi prediction model on each 100 base increment (i.e., bin) improved the true positive to false positive ratio compared to the original transcriptome (produced by StringTie) while simultaneously decreasing the number of false negatives in favor of true negatives (Supplementary Fig. 3).

We next compared aptardi to TAPAS[45]—identified by Chen et al.[35] as the top performer for characterizing APA from RNA-Seq—using the same 100 base distance cutoff to define true positives (i.e., if a TAPAS prediction was within 100 bases of any true polyA site in the HBR PolyA-Seq data it was considered a true positive, otherwise it was a false positive). The aptardi pipeline identified the 3′ termini correctly for 62,688 transcripts compared to 3′ termini of 22,804 transcripts using TAPAS. Although the number of transcripts with a false positive 3′ terminus was higher in the aptardi modified transcriptome compared with TAPAS[45] owing to annotations from the original transcriptome, the positive predictive value was higher for the aptardi modified transcriptome because it added many more true positive than false positive 3′ termini to the original transcriptome (aptardi modified transcriptome = 0.44, TAPAS = 0.31). Finally, the aptardi pipeline captured more unique true polyA sites (as identified by the HBR PolyA-Seq data) compared with both TAPAS and the original transcriptome (aptardi modified transcriptome = 29,327, TAPAS = 25,180, original transcriptome = 23,685). Similar results were achieved when adjusting the base distance cutoff for true positives and/or utilizing the PolyASite 2.0 and PolyA_DB databases to define true polyA sites (Supplementary Table 2).

A final comparison was made to APARENT[60], which utilizes only DNA sequence to make predictions. Its positive predictive value was lower than that for aptardi for all databases at all base distance cutoffs (Supplementary Table 2). Notably, although

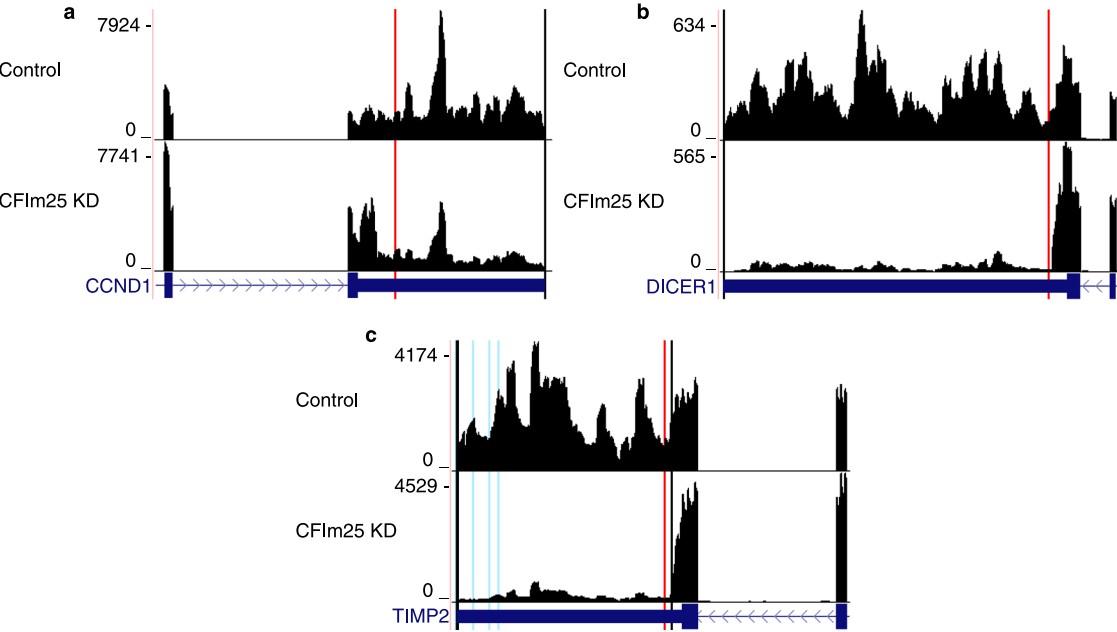

**Fig. 5 Aptardi displays sample-specific sensitivity when annotating transcription stop sites.** RNA-sequencing (RNA-Seq) read densities for **a** *CCND1*, **b** *DICER1*, and **c** *TIMP2* after control (Control) siRNA treatment and CFIm25 knockdown (KD) in HeLa cells. Numbers on *y* axis indicate RNA-Seq read coverage. After knockdown, each gene preferentially expresses a proximal alternative polyadenylation (APA) site compared to under control conditions. Transcript structures shown are from RefSeq annotation (dark blue), where boxes and lines indicate exons and introns, respectively. Black vertical lines indicate transcript stop sites identified in the original transcriptome, red vertical lines indicate transcript stop sites only identified in the aptardi modified transcriptome and that match the original study's findings, and blue vertical lines indicate transcript stop sites only identified in the aptardi modified transcriptome that are not described in the original study. Graphics were generating using the UCSC Genome Browser (https://genome.ucsc.edu/) using the hg38 human genome assembly.

APARENT annotated more true polyA sites as defined in the PolyASite 2.0 and PolyA_DB databases compared with all the other methods (likely owing to its high number of overall predictions), this increase was not observed when being compared with aptardi and using the HBR PolyA-Seq data to define the true polyA sites, which likely more accurately represents the polyA sites being expressed in the corresponding HBR RNA-Seq data (Supplementary Table 2).

**Aptardi identifies sample-specific transcripts missed by current transcriptome reconstruction methods.** We next sought to ascertain if aptardi could identify APA transcripts observed in a previous study where differential APA expression was induced by knocking down the cleavage and polyadenylation machinery CFIm25[61]. In this study, the authors experimentally confirmed expression of short APA transcript isoforms after CFIm25 knockdown for three genes capable of undergoing APA[62,63]— *CCND1*, *DICER1*, and *TIMP2*—and used DaPars[43] to computationally estimate the locations of polyA sites.

For each the control and knockdown RNA-Seq data set, the aptardi modified transcriptome was compared with the original transcriptome, which contained both Ensembl annotations and sample-specific expressed transcripts identified through StringTie reconstruction. In the control RNA-Seq data set, neither aptardi nor the original transcriptome identified a shorter APA transcript for *CCND1* in agreement with the original study design; in the knockdown treatment RNA-Seq data set, only aptardi recapitulated the short APA isoform (Fig. 5a), demonstrating its sensitivity to sample-specific data and its ability to improve upon current annotation methods. Likewise, only aptardi identified the proximal APA transcript for *DICER1* (Fig. 5b). For *TIMP2*, multiple transcript isoforms are annotated in Ensembl[59], and StringTie[22] retained all these transcripts in its reconstruction. In contrast,

aptardi annotated a short APA transcript only in the treatment consistent with Masamha et al.[61], again demonstrating its sample-specific sensitivity (Fig. 5c). Finally, the locations of the proximal transcripts for these genes identified by aptardi were similar to the original study (*CCND1*: aptardi = chr11: 69,651,917, original study = chr11: 69,651,578; *DICER1*: aptardi = chr11: 95,090,264, original study = chr11: 95,090,400; *TIMP2*: aptardi = chr17: 78,855,465, original study = chr17: 78,855,601).

**Comparison of aptardi predictions across mouse tissues.** The above study demonstrated aptardi's ability to differentiate polyA sites across samples; however, we extended this analysis by comparing different mouse tissues—namely liver and brain—to mimic subtle differences in polyA sites. Brain and liver RNA-Seq data were procured from Li et al.[64], true polyA sites were derived again from PolyA-Seq data that were specific to brain and liver, and the mouse reference mm10/GRCm38 genome was used for DNA sequence for both tissues (see Mouse tissue analysis in Methods for more details).

For the PolyA-Seq data, we identified polyA sites that (1) were unique to either brain or liver, (2) were within the 3′ modified terminal exon of a transcript in the StringTie generated original transcriptome (i.e., made available to aptardi) for both tissues, and (3) did not coincide with a previously annotated Ensembl transcript end. Using these restrictions, we were able to focus on unannotated polyA sites that differed across tissues but were associated with genes/transcripts expressed in both tissues. This resulted in 756 unannotated brain-specific sites and 1529 unannotated liver-specific sites. The StringTie pipeline was able to capture three of the unannotated brain-specific sites and four of the unannotated liver-specific sites. Including the aptardi prediction model (built from HBR) in the transcript discovery pipeline added 26 unannotated brain-specific sites (fold

increase = 9) and 69 unannotated liver-specific sites (fold increase = 17) Furthermore, only nine of these 26 additional unannotated brain-specific polyA sites were also added to the liver data using aptardi (i.e., aptardi did not distinguish the brain from liver) and only 25 of these 69 unannotated liver-specific polyA sites were identified by aptardi in the brain data.

**Evaluation of the use of aptardi in a differential expression pipeline.** The influence of aptardi on differential expression analysis was evaluated using the BNLx and SHR rat brain data sets by evaluating transcripts identified as differentially expressed between strains with the aptardi modified transcriptome ($p$ value ≤ 0.001) but not the original transcriptome derived from the StringTie/Ensembl pipeline ($p$ value ≥ 0.001). Note that since aptardi incorporates transcripts into annotation, expression levels of existing transcripts can also change, i.e., transcripts present in both the aptardi modified transcriptome and the original transcriptome may be identified as differentially expressed in one and not the other. A total of 1166 out of 32,348 transcripts and 918 out of 28,329 transcripts expressed above background were differentially expressed ($p$ value ≤ 0.001) using the aptardi modified transcriptome and original transcriptome, respectively. A total of 40 transcripts that could be associated with an Ensembl gene symbol were differentially expressed in the aptardi modified transcriptome but not in the original transcriptome although they had identical structures, including 3′-ends, in both (i.e., original transcriptome transcripts). Furthermore, 54 aptardi transcripts

that could be associated with a gene symbol were differentially expressed and NOT measured/identified in the original transcriptome (Supplementary Data 1). The RNA-Seq read coverage for six of these aptardi transcripts are depicted in Fig. 6. For *Unc79* (Fig. 6a), *Sf3b1* (Fig. 6b), *Ptn* (Fig. 6c), and *Ap3b1* (Fig. 6d), the original transcript was differentially expressed in the aptardi modified transcriptome but not the original transcriptome, and for *Zdhhc22* (Fig. 6e) and *RGD1559441* (Fig. 6f) the aptardi transcript was differentially expressed. The RNA-Seq read coverage across these genes support the presence of the aptardi transcripts and differential expression of the various isoforms between strains. Moreover, these results demonstrate that aptardi is capable of identifying both shortening and lengthening events—e.g., four of the six genes were annotated with a shorter transcript by aptardi and two of the six a longer one—as well as identifying isoforms across a broad range of RNA-Seq coverage depths; the peak coverage value for each gene ranged from ~200 to 8000.

**Discussion**

Aptardi leverages the information afforded by both DNA sequence and short-read RNA-Seq to accurately annotate the polyA sites of expressed transcripts in a biological sample. We first established the applicability of aptardi by showing that (1) a prediction model derived from a single data set performed well on data sets that differ on technical issues and even species, (2) the process of training the prediction model is generalizable across different types of RNA-Seq data/DNA sequence, and (3) the algorithm is not prone to

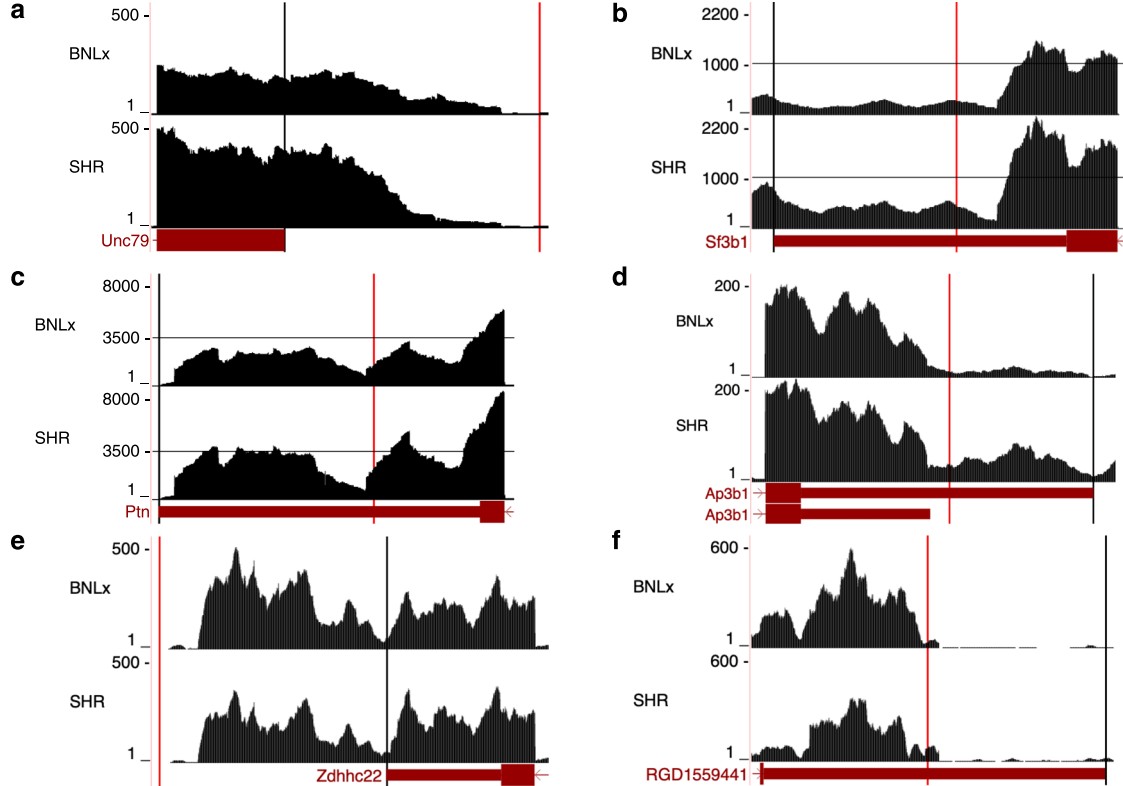

**Fig. 6 Incorporation of aptardi into differential expression analyses.** RNA-sequencing (RNA-Seq) read densities for six genes in BNLx and SHR inbred rat strains. Numbers on *y* axis indicate RNA-Seq read coverage. Read coverage represents the aggregate of three biological samples for each strain. Transcript structures shown are from Ensembl annotation (dark red), where boxes and lines indicate exons and introns, respectively. Black vertical lines denote transcript stop sites identified in the original transcriptome derived using StringTie, and red vertical lines indicate transcript stop sites identified in the aptardi modified transcriptome only. No transcripts were identified as differentially expressed between strains in the original transcriptome (p > 0.001), but at least one differentially expressed transcript for each gene was identified in the aptardi modified transcriptome ($p$ ≤ 0.001). For **a** *Unc79*, **b** *Sf3b1*, **c** *Ptn*, and **d** *Ap3b1* the original transcript isoform (black line) was differentially expressed in the aptardi modified transcriptome, and for **e** *Zdhhc22* and **f** *RGD1559441* the aptardi transcript was differentially expressed (red line). Graphics were generating using the UCSC Genome Browser (https://genome.ucsc.edu/) using the rn6 rat genome assembly.

overfitting. Namely, we showed that the aptardi prediction model provided for users (built from the HBR data set) performs equally well on RNA-Seq data sets derived from different library preparations, organisms, and with different RNA-sequencing depths. We note that aptardi performed modestly worse on the BNLx and SHR data sets and hypothesize this is because the true polyA sites were derived from the Sprague Dawley rat instead of the specific rat strain. This is supported by the fact that prediction models built from the BNLx and SHR data sets and tested on these same data sets performed similarly to the aptardi prediction model built on the HBR data set (Supplementary Fig. 2). However, we cannot rule out the possibility that this is owing to the unstranded RNA-Seq for these data sets and/or different species. Moreover, the comparable results when models were built and evaluated on a single data set versus models applied to different data sets from those used to train the model suggest the data processing pipeline/prediction models are generalizable. Finally, the similarity of the AP estimates between training and testing sets demonstrate that aptardi is not prone to overfitting. Overall, these results indicate aptardi can be broadly used.

We next established that incorporating aptardi into current transcriptome reconstruction methods improves annotation of 3′-ends. This was done by comparing the aptardi modified transcriptome to the original transcriptome assembled using the power of both existing annotation via Ensembl and taking into consideration RNA-Seq coverage via StringTie. Adding aptardi transcripts increased the number of unique true polyA sites captured by the transcriptome and furthermore increased the ratio of true positive to false positive termini compared with the original transcriptome. Aptardi also outperformed TAPAS in these respects, and TAPAS was previously identified as the top performer for identifying polyA sites from RNA-Seq[35]. Likewise, aptardi produced greater positive predictive values compared to the deep learning algorithm APARENT that utilizes DNA sequence to make predictions and, furthermore, annotated more polyA sites at 100 and 50 base distance cutoffs and nearly the same at a base distance cutoff of 25 despite making many fewer predictions.

Applying aptardi in control and CFIm25 knockdown RNA-Seq data demonstrated its (1) sensitivity to sample-specific expression, (2) ability to identify both shortening and lengthening APA events, (3) competence across a broad range of RNA-Seq coverage depths, and (4) ability to improve upon current reconstruction methods. Of interest, in the control for *TIMP2*, aptardi identified several 3′-ends close to the annotated distal transcript that were not noted in the original study[61]. The RNA-Seq from the control sample displays uneven coverage in this region, meaning aptardi may have uncovered additional, previously unknown isoforms (Fig. 5c). Of note, the library preparation for these RNA-Seq data were unstranded (unlike the data used to generate aptardi), further supporting aptardi's broad applicability. These results also highlight potential weaknesses of aptardi. For instance, aptardi incorporates a single 3′-end into multiple transcripts for each gene listed because it does not distinguish transcripts overlapping the same genomic region. This may be somewhat mitigated by curating a more-selective input transcriptome. Here the entire Ensembl annotation was provided, which includes many "pseudo" transcripts (e.g., retained introns, nonsense-mediated decay, and isoforms only identified computationally), and these transcripts often overlap manually identified mRNAs. Second, aptardi will add polyA sites for a transcript regardless of if another transcript isoform from the same gene already has a transcript stop site at the given location, as is the case for *TIMP2*. As a result, it is possible that aptardi incorporates a transcript stop site belonging to a different transcript.

The sample-specific sensitivity of aptardi was also evaluated using RNA-Seq from brain and liver mouse tissues and evaluating its ability to identify polyA sites unique to each tissue. Aptardi successfully annotated tissue-specific polyA sites not previously reported in Ensembl annotation or discovered by StringTie reconstruction. Furthermore, aptardi was able to provide a significant fold increase in polyA site annotation compared with StringTie, exemplifying its sensitivity. Although the overall number of polyA sites detected was modest, this was likely owing to the low sequencing depths of the samples (Supplementary Table 3). As a result, many polyA sites reported in the PolyA-Seq data—which was sequenced at much greater depths—were likely not detectable here. Indeed, the reference mouse Ensembl genome annotated a greater number of both overall polyA sites (brain = 14,630, liver = 13,209) and tissue-specific polyA sites (brain = 5144, liver = 3668) compared with the StringTie pipeline (overall: brain = 11,285, liver = 9572; tissue-specific: brain = 3468, liver = 2331). StringTie removes guide transcripts (in this case Ensembl transcripts) below a minimum coverage threshold (one when using default settings), suggesting many PolyA-Seq polyA sites—including those in the 3′ modified terminal exons that aptardi did not annotate—were not detectable here.

We further examined how incorporating aptardi into downstream transcriptome analyses such as differential expression may alter the interpretation of results. We found that multiple isoforms —some of which were already present in the original transcriptome and some of which were transcripts identified by aptardi—were differentially expressed between BNLx and SHR recombinant inbred rats only when using the aptardi modified transcriptome. Some of these transcripts are derived from genes that have also been implicated in phenotypes related to the SHR rat, such as greater sensitivity to addictive drugs[65] and increased voluntary ethanol consumption[66]. For instance, expression of *Ptn*—which has verified APA sites[67]—is modulated by amphetamine in rat nucleus accumbens[68]. Furthermore, *Unc79* knockout mice displayed hypersensitivity to ethanol, e.g., increased preference for and consumption of alcohol[66]. Undoubtedly, further investigation is needed to elucidate the role of *Ptn and Unc79* APA on addiction phenotypes, but these preliminary results demonstrate how aptardi may help unravel the genetic architecture of complex diseases. Of note, although Ensembl annotation provides two transcript isoforms for *Ap3b1*, StringTie assembly resulted in inclusion of the longer isoform only, highlighting the difficulty of current assembly methods for identifying 3′-ends of transcripts embedded in a longer version. However, aptardi identified a shorter transcript within 100 bases of the original Ensembl annotation for the shorter transcript that is likewise supported by the RNA-Seq data (Fig. 6d).

Also of note, aptardi is easily integrable into existing analyses pipelines. For instance, unlike current supplemental methods designed for APA detection from RNA-Seq, no additional data manipulation is required prior to running the program. The input files are readily available (e.g., reference genome and reference transcriptome) or already generated during the course of transcriptomic analysis (e.g., RNA-Seq data and a reconstructed transcriptome). The output GTF file can be used in the same manner as other annotation files (e.g., those accessed via Ensembl or generated via a transcriptome assembler such as StringTie). Moreover, the program can be seamlessly integrated into a single operation with upstream transcriptome assembly and downstream analyses (e.g. quantitation) via piping to make for streamlined analysis. Finally, there is also the option of constructing a prediction model using the aptardi architecture, which increases the breadth of its applicability to diverse data sources.

An additional perceived limitation may be that aptardi makes predictions on 100 base bins (for positive predictions, it annotates

the transcript stop site as the 3′ most base in the given 100 base bin). However, this concern is partially mitigated because the precise location of polyA sites can "wiggle" by up to 30 nucleotides for what is considered a single isoform, i.e., not an APA event[50], and as such researchers often group polyA sites within 30 bases into a single site[46]. Furthermore, few 100 base bins contained multiple polyA sites (Supplementary Table 4). Another potential limitation is that upstream exons are not subjected to aptardi analysis, effectively eliminating the possibility of identifying coding APA; fortunately, the vast majority of APA sites do not result in a change within the protein-coding region[2]. Finally, the manually engineered DNA sequence features may not apply to taxa outside of mammals[50] and will require further research.

Transcriptome profiling is one of the most-utilized approaches for investigating human diseases at the molecular level, yielding important insights into many pathologies. A prerequisite for these studies is a representative transcriptome map. Aptardi incorporates APA transcripts to produce a more-accurate transcriptome map, thereby enabling future research into the role of APA transcripts—as well as other transcripts unencumbered by convoluted annotation with APA transcripts–in human health and disease.

## Methods

**Aptardi design**. The overall goal of aptardi is to accurately identify the polyA sites of expressed transcripts in a given biological sample. Specifically, aptardi analyzes the modified 3′-terminal exon (see the Transcript processing section below for details on 3′-terminal exon modification) of previously annotated transcripts and, using relevant RNA-Seq data and DNA sequence in a machine learning environment, identifies locations of expressed polyA sites in the region. Aptardi then annotates the 3′-termini to match these locations and outputs these transcript structures to the transcriptome (in GTF format) that can be easily incorporated into downstream analyses. Note that aptardi does not evaluate the intron chain structure of transcripts, i.e., it only examines the modified 3′ terminal exon of each transcript structure and alters the 3′ terminus location(s) accordingly. Also, note that aptardi outputs all original transcript structures from the original transcriptome in addition to transcripts identified through its analysis, i.e., the program only adds transcripts.

**Data sets**. A total of five unique data sets, hereafter referred to as HBR, 2nd HBR, UHR, BNLx, and SHR, were subjected to aptardi's machine learning pipeline. In addition to RNA-Seq measurements, each data set required DNA sequence, a transcriptome, and—since each was used to build a machine learning model—a "gold standard" data source providing locations of expressed, i.e., "true" polyA sites. HBR, 2nd HBR, and UHR are well-established RNA reference samples from the MAQC/SEQC consortium[54] (see RNA-sequencing data sets for more details). BNLx and SHR represent two inbred rat strains: the congenic Brown Norway strain with polydactyly-luxate syndrome (BNLx/Cub) and the spontaneous hypertensive rat strain (SHR/OlaIpcv), respectively.

**DNA sequence data sets**. For BNLx and SHR, strain-specific genomes were generated from the rn6/Rnor_6.0 version of the rat genome[69] and are publicly available on the PhenoGen website. The human reference genome (hg38/GRCh38), accessed via the UCSC Genome Browser[70], was utilized for the HBR, 2nd HBR, and UHR DNA sequence data sets.

**RNA-sequencing data sets**. The HBR and 2nd HBR RNA-Seq data sets were derived from the Human Brain Reference (multiple brain regions of 12 donors, Ambion, p/n AM6050), and the UHR RNA-Seq data set was derived from the Universal Human Reference (10 pooled cancer lines, Stratagene, p/n 740000). Each of these data sets were accessed from the Sequence Read Archive (SRA) using the SRA Toolkit (v.2.8.2) as publicly available data (HBR[55]: Accession: PRJNA510978, SRA runs: SRR8360036-37; 2nd HBR and UHR[56]: Accession: PRJNA362835, 2nd HBR SRA runs: SRR5236425-30, UHR SRA runs: SRR5236455-60; BNLx and SHR[71]: Accession: GSE166117, BNLx: GSM5061950-52, SHR: GSM5061947-49). In brief, all libraries were generated with the TruSeq stranded (HBR, 2nd HBR, UHR) or unstranded (BNLx and SHR) mRNA sample preparation kit (Illumina), sequenced on a HiSeq2500 Instrument (Illumina), and sequencing results processed to FASTQ files. The HBR RNA-Seq data set originated from 1 μg RNA starting material, whereas 100 ng input was used for 2nd HBR and UHR RNA-Seq data sets. For more detailed descriptions on these publicly available data, see Palomares et al.[55] (HBR) and Schuierer et al.[56] (2nd HBR and UHR). For BNLx and SHR, RNA-seq libraries prepared from the polyA+ fraction were constructed using the Illumina TruSeq RNA Sample Preparation kit from 1 μg of brain RNA in accordance with the manufacturer's instructions. Four μL of a 1:100 dilution of either ERCC Spike-In Mix 1 or Mix 2 (ThermoFisher Scientific) were added to each extracted RNA sample. An

Agilent Technologies Bioanalyzer 2100 (Agilent Technologies) was utilized to assess sequencing library quality. RNA samples from three biological replicates per strain were processed and sequenced[71]. All reads were paired-end but differed in read length (HBR, BNLx, and SHR: 2 × 100, 2nd HBR and UHR: 2 × 75). Individual FASTQ files were assessed for quality using FastQC (v.0.11.4, https://www.bioinformatics.babraham.ac.uk/projects/fastqc/) and, if necessary, reads were trimmed with cutadapt[72] (v.1.9.1). For the purpose of read coverage used by the aptardi algorithm, reads from technical replicates (HBR = 2, 2nd HBR = 3, UHR = 3) or biological replicates (BNLx = 3, SHR = 3) were concatenated and aligned to their respective genomes (see DNA sequence data sets section for more details) using HISAT2[73] (v.2.1.0) with the–rna-strandness (when appropriate) and–dta options specified as recommended for transcriptome assembly with StringTie[22] (see Transcriptome data sets below for more details) and otherwise default arguments (see Supplementary Table 5 for alignment results). After alignment, SAMtools[74] (v.1.9) was used to remove unmapped reads and convert the output to a sorted Binary Alignment Map (BAM) file required as input by aptardi.

**True polyadenylation sites data sets**. True polyA sites (i.e., labels for machine learning) were taken from Derti et al.[46] for all data sets. Namely, total RNA from the same UHR and HBR RNA reference samples, as well as brain total RNA from the Sprague Dawley rat (Zyagen, p/n RR-201), were subjected to PolyA-Seq analysis to identify the genomic locations of expressed polyA sites in each sample (for more information, see Derti et al.[46]). High-quality filtered polyA sites from each RNA sample were accessed using the UCSC Table Browser[75], and liftOver[76] (from the UCSC Genome Browser Group) was used to convert the genomic coordinates to the most recent human genome assembly (hg38/GRCh38) for the HBR and UHR samples, or rat genome assembly (rn6/Rnor_6.0) for the rat brain sample. PolyA sites identified in the HBR and UHR RNA reference samples were used for the corresponding HBR, 2nd HBR and UHR data sets, and those identified in the rat brain were used for both BNLx and SHR. Derti et al.[46] uploaded technical replicates to UCSC Table Browser for each RNA sample; however, as polyA sites within 30 bases were clustered into the single site with greatest expression, and since this was done separately for each data set, we utilized only a single data set for each sample, i.e., technical replicates were not combined.

**Original transcriptome generation**. StringTie[22] (v.1.3.5) was used to reconstruct the transcriptome expressed in each data set from their RNA-Seq data, hereafter referred to as the original transcriptome. Ensembl[59] (v.99) reference annotation from the respective species was provided to guide the StringTie reconstruction. We note that a user can simply use reference annotation directly, i.e., Ensembl annotation, in lieu of performing transcriptome assembly that takes into account expression, i.e., StringTie. If the RNA-Seq data were stranded, the read orientation was specified as an argument to StringTie. Transcript structures from scaffold chromosomes and unstranded contigs, if present, were removed.

**The data processing pipeline**

*Transcript processing*. Using the original transcriptome, the 3′ terminal exons of transcripts were isolated. Each transcript's 3′ terminal exon was extended 10,000 bases plus two times the bin size (i.e., 10,200 bases for the default 100 base bin size) similar to what has been done previously[43,44]. Extensions overlapping any neighboring transcripts (on the same strand) were shortened to remove the overlap. RNA-Seq coverage at single-nucleotide resolution was obtained via bedtools genomecov (BEDtools[77]; v2.29.2) and, similar to the criteria employed by Ye et al.[44] and Miura et al.[78], used to refine each of these 3′ terminal exons, hereafter referred to as modified 3′ terminal exons (see Supplementary Methods for more details.) The refinement step either shortened the extended 3′ terminal exon or kept it the same length to give the modified 3′ terminal exon.

*Feature extraction*. Features were engineered in 100 base increments along the modified 3′ terminal exon, referred to hereafter as bins. For each bin, a total of 27 features were engineered and can be broadly classified as being derived from DNA sequence or RNA-Seq data. In both cases, information from the local environment, i.e., the 100 bases upstream and downstream the bin, as well as the bin itself, (300 bases total) was used.

*DNA sequence features*. The choice of DNA sequence features was made through a combination of an exhaustive literature review[25,50,79–86] and evaluation of other algorithms that use DNA sequence to predict polyA sites[52,53,87,88]. Perhaps the most well-known indicator of polyadenylation is the polyadenylation signal (PAS), a conserved hexamer located ~10–35 nucleotides upstream the polyA site. Overrepresented sequences of DNA, or DNA sequence elements, also influence polyadenylation, and the location of these sequences are often described relative to the PAS. As such, DNA sequence features are engineered by first identifying the presence of several known PAS's. Specifically, for each bin, a six base sliding window scanned a predefined region to detect the presence or absence (binary indicator of 1/−1) of (1) the canonical PAS (AATAAA), (2) its major variant (ATTAAA), (3) a second common variant (AGTAAA), and (4) any one of nine other minor variants (AAGAAA, AAAAAG, AATACA, TATAAA, GATAAA, AATATA, CATAAA, AATAGA[25,50,79,80])

for four total PAS features. Subsequently, regions relative to the PAS (if present, otherwise predefined regions relative to the current bin) were likewise scanned using a sliding window approach to determine frequency of the following known DNA sequence elements: (1) a G-rich region downstream the PAS, (2) a downstream region near the PAS enriched in TTT, (3) a downstream region near the PAS enriched in GT/TG, and (4) a downstream region near the PAS enriched in GTGT/TGTG, (5) a T-rich region immediately downstream of the PAS, (6) a T-rich region upstream the PAS, (7) a TGTA/TATA-rich region upstream the PAS, and (8) a AT-rich region upstream and downstream the PAS[25,50,81–86] for an additional eight features (12 DNA sequence features total). If the frequency of the given DNA sequence element was above an enrichment threshold, the feature was encoded 1, otherwise −1. (See Supplementary Methods for more details.)

*RNA-sequencing features*. From the RNA-Seq data, coverage at single-nucleotide resolution was determined using BEDtools[77] (v2.29.2). The approach for designing RNA-Seq features was to exploit localized fluctuations in RNA read coverage similar to that implemented by tools designed for APA-specific analysis from RNA-Seq data[43–45]. Intuitively, upstream but in close proximity to the end of a transcript, coverage is expected to begin to decrease gradually until its end. As a result, changes in expression were utilized when designing RNA-Seq features in two scenarios: (1) intra- and (2) inter-bin. In both cases, three regions were defined: an upstream region, a middle region, and a downstream region (Supplementary Fig. 4). Changes in expression between these regions were quantified using various mathematical combinations of coverage values in each region to generate 14 unique features (see Supplementary Methods for more details). To account for local variability in RNA-Seq coverage, median coverage values in each region were used.

A final feature was derived from the original transcriptome. If the 3′ base of any annotated transcript from the original reconstruction was located within a bin, this feature was encoded 1, otherwise −1. Supplementary Fig. 5 summarizes the data processing pipeline prior to machine learning.

**Building aptardi**. The machine learning task is two class classification (polyA site or no polyA site) of each 100 base bin. Supervised learning was used where labels for training were provided from the polyA sites data sets (see PolyA sites data sets for more details). A bidirectional long short term memory recurrent neural network (biLSTM)[89–92] was implemented using the Keras (v.2.3.1) wrapper for TensorFlow (v.2.0.0). This machine learning paradigm was chosen because of its design to analyze sequential data, i.e., it takes into account all the 100 base bins of a given transcript when learning model parameters for each individual bin. Each direction of the biLSTM consisted of 20 nodes (40 total), and this layer was followed by a fully connected dense layer with a sigmoid activation function that outputs a probability value. Of note, the biLSTM outperformed traditional classifiers such as Random Forest (RF) and Support Vector Machine (SVM) models (biLSTM: AP = 0.58, F-measure = 0.52; RF: AP = 0.42, F-measure = 0.39; SVM: AP = 0.37, F-measure = 0.33; numbers shown are on the testing set using the HBR data set).

**Training aptardi**. To prevent duplicate bins, overlapping modified 3′ terminal exons were merged prior to training. In addition, all merged modified 3′ terminal exons were masked to a length of 300 bins (30,000 bases total) to generate equal lengths, which is required for the sequential data. A total of 778,166 bins were present, of which 42,977 possessed a polyA site out of 94,322 polyA sites annotated by the PolyA-Seq data (for the HBR data set). Merged modified 3′ terminal exons were split into 60/20/20 training, validation, testing sets, respectively. Quantitative measures were standardized using the training set, and the training set was used to build the model in 25 epochs. Owing to the high imbalance of the data, class weights were used during training. Model weights were optimized using a binary cross entropy loss function and Adam[93] optimizer. Precision and recall metrics on the training and validation sets were monitored during training to prevent over-fitting, and the model that produced the minimum loss was kept. For evaluation purposes (see Results), individual prediction models were generated from each of the five data sets.

**Evaluating aptardi**. Precision, recall, and F-measure at the default probability threshold (0.5) were used to evaluate model performance defined as follows:

$$P = \frac{T_p}{T_p + F_p} \quad (1)$$

$$R = \frac{T_p}{T_p + F_n} \quad (2)$$

$$F = 2 * \frac{(P * R)}{(P + R)} \quad (3)$$

where $T_p$ = true positive, $F_p$ = false positive, $F_n$ = false negative, P = precision, R = recall, and F = F-measure

To generalize model performance over the range of probability thresholds, AP was used in place of the receiver operating curve owing to the highly imbalanced nature of the data[94] (far fewer bins with polyA sites than bins without a polyA site):

$$AP = \sum_n (R_n - R_{n-1}) P_n \quad (4)$$

where AP = average precision and $R_n$ and $P_n$ are the precision and recall at the $n$th threshold, respectively.

**Integrating aptardi results with the original transcriptome**. For 100 base bins where a polyA site is predicted, transcript structures are annotated to the 3′ most base position unless either (1) the input transcript's stop site is already in the region or (2) the 3′ most base position is within 100 bases of the input transcript's stop site. Any aptardi transcript structures were added to the original transcriptome, and this aptardi modified transcriptome was outputted as a GTF file.

**Choice of bin size**. To assess the impact of bin size, 25, 50, and 150 base bins were used to train the aptardi prediction model on the HBR data set in an otherwise identical manner to the original 100 base bin. The AP on the testing set was 0.41, 0.51, and 0.61 for the 25, 50, and 150 base bins, respectively, compared with 0.58 for the original 100 base bin. In addition, the F-measure on the testing set was 0.35, 0.46, and 0.52, respectively, compared with 0.52 for the for the original 100 base bin. Based on these results, 100 base bins were utilized to build the pre-existing aptardi prediction model. Higher resolution bin sizes may be appropriate depending on the data set (e.g., RNA-Seq library preparation), species, etc. and bin size can be specified as a parameter when using the aptardi pipeline.

**Software**. A user has the option of using the pre-existing aptardi prediction model or building a prediction model (if a reliable true polyA sites data set is available). The pre-built model, i.e., the aptardi prediction model, provided on the aptardi GitHub repository (https://github.com/luskry/aptardi) was generated from the HBR data set[55]. Several other algorithm options are available.

**TAPAS analysis**. TAPAS (Tool for Alternative Polyadenylation site AnalysiS) predicts the locations of polyA sites from RNA-Seq and reference annotation (genome or transcriptome)[45]. Its performance was evaluated on HBR, specifically using the HBR RNA-Seq data and StringTie/Ensembl original transcriptome. As TAPAS makes predictions on noncoding sequence coordinates of transcript models (i.e., 3′-untranslated region), the 3′ terminal exon of each transcript was provided as the noncoding region, and default arguments were used. The TAPAS polyA site prediction program (APA_sites_detection) was used here and accessed via its GitHub repository (https://github.com/arefeen/TAPAS).

**APARENT analysis**. APARENT (APA REgression NeT) predicts the locations of polyA sites from DNA sequence[60]. Its performance was evaluated using the human reference genome (hg38/GRCh38) and transcript structures generated from the HBR StringTie/Ensembl original transcriptome. Specifically, similar to that for aptardi, the 3′ terminal exon of each of the 113,923 transcript models in the original transcriptome were extracted. Since APARENT requires DNA sequences to be ≥205 nucleotides and ≤10,000 nucleotides, 20,658 3′ terminal exons were removed from the analysis. For the remaining 93,265 3′ terminal exons, the DNA sequence was extracted using BEDtools (v2.29.2). For 3′ terminal exons on the negative strand, the reverse complement sequence was used. The APARENT model (aparent_large_lessdropout_all_libs_no_sampleweights.h5) from its GitHub repository (https://github.com/johli/aparent) was used to predict the locations of polyA sites using default parameters.

**CFIm25 knockdown analysis**. RNA-Seq from HeLa cells and RNA-Seq after RNA interference on HeLa cells was used to generate the control RNA-Seq data set and the treatment CFIm25 knockdown RNA-Seq data set that induces APA switching, respectively. The RNA-Seq data sets were accessed from SRA as publicly available data (Accession: PRJNA182153, control SRA run: SRR1238549, CFIm25 knockdown SRA run: SRR1238551). The RNA-Seq library preparation was unstranded, and 100 base paired-end reads were sequenced on an Illumina HiSeq 2000 instrument (see Masamha et al.[61] for more details). Reads were processed in a manner identical to all other data sets to produce a sorted BAM file (see Supplementary Table 6 for alignment results). An original transcriptome using StringTie/Ensembl (without the read orientation argument) and an aptardi modified transcriptome were generated for each RNA-Seq data set (four total). The aptardi prediction model produced using the HBR data set was used to generate the aptardi modified transcriptomes.

**Mouse tissue analysis**. RNA-Seq from mouse brain and liver tissues were taken from Li et al.[64] and accessed from SRA as publicly available data (Accession: PRJNA375882, brain SRA runs: SRR5273637 and SRR5273673, liver SRA runs: SRR5273636 and SRR5273672). The two SRA runs per tissue represent technical replicates from one biological sample—namely 6-week old female C57BL/6JJcl mice. In brief, libraries were constructed using polyA selection and the Illumina TruSeq RNA-Seq library protocol

and sequenced using an Illumina HiScan platform to generate 100 base, paired end, and unstranded reads. Reads were trimmed for quality and adapter content with Trimmomatic[95] (v.0.39), and the two technical replicates per tissue were aligned together to the mm10/GRCm38 genome using HISAT2 (v.2.1.0; see Supplementary Table 3 for alignment results). An original transcriptome was generated using StringTie (v.1.3.5) and the mouse Ensembl (v.102) annotation as a guide and otherwise default settings. BALB/c mouse brain and liver PolyA-Seq data from Derti et al. were used to define the tissue-specific true polyA sites. Namely, high-quality filtered true polyA sites were accessed using the UCSC Table Browser, and liftOver (from the UCSC Genome Browser Group) was used to convert the genomic coordinates to the most recent mouse assembly. Tissue-specific polyA sites in brain and liver were defined as polyA sites in the given tissue PolyA-Seq data set not within 100 bases of any polyA site in the other tissue PolyA-Seq data set. The mouse reference genome (mm10/GRCm38), accessed via the UCSC Genome Browser was used for DNA sequence for both tissues. Comparisons between the polyA sites from the given PolyA-Seq data and transcriptome were done by defining that a transcript annotated a polyA site if its 3′ terminus was within 100 bases of the site.

**Rat differential expression analysis**. To generate a single transcriptome representing both rat strains, their genome-aligned RNA-Seq data were merged using SAMtools[74] (v.1.9) followed by the production of an original transcriptome using the merged RNA-Seq data set and StringTie/Ensembl (without the read orientation argument). The aptardi modified transcriptome was produced using the original transcriptome as input, along with the merged RNA-Seq data, the rn6/Rnor_6 DNA sequence (accessed via the UCSC Genome Browser[70]), and the aptardi prediction model built from HBR. RSEM[96] (v.1.2.31) was used to estimate the abundances of the isoforms identified within each transcriptome (the original transcriptome and aptardi modified transcriptome). Prior to quantitation, transcripts from scaffold chromosomes and unstranded contigs were removed from both transcriptomes. Isoform level expression estimates were determined for each biological sample (BNLx = 3, SHR = 3). Isoforms without at least 50 counts in two of the three biological replicates for at least one strain were removed, and differential expression between the two strains (with BNLx as reference) was evaluated using DESeq2[97] (v.1.28.0) for the remaining set of isoforms in each transcriptome (Supplementary Fig. 6 summarizes these analysis steps). A significance threshold of 0.001 was applied to the unadjusted $p$ values to allow for comparisons across the two data sets (original transcriptome and aptardi modified transcriptome) that differ in the number of transcripts tested.

**Reporting summary**. Further information on research design is available in the Nature Research Reporting Summary linked to this article.

## Data availability

All data are publicly available. The genomic sequence data that support the findings of this study are available on the UCSC Genome Browser (human: hg38/GRCh38, rat: rn6/Rnor_6.0, mouse: mm10/GRCm38, http://genome.ucsc.edu/) and PhenoGen (BNLx: BNLx/CubPrin, SHR: SHR/OlaIpcvPrin, https://phenogen.org/). The polyadenylation sites from PolyA-Seq data that support the findings of this study are available on the UCSC Table Browser (https://genome.ucsc.edu/). The polyadenylation sites from PolyA_DB and PolyASite 2.0 that support the findings of this study are available on their respective websites (PolyA_DB: https://exon.apps.wistar.org/PolyA_DB/v3/, PolyASite 2.0: https://polyasite.unibas.ch/). The RNA-Sequencing data that support the finding of this study are available on the NCBI SRA (Human Brain Reference RNA sequencing: accession: PRJNA510978, SRA runs: SRR5236425-30; 2nd Human Brain Reference and Universal Human Reference RNA sequencing: accession: PRJNA362835, 2nd HBR SRA runs: SRR5236425-30, UHR SRA runs: SRR5236455-60; Control vs CFIm25 knockdown RNA sequencing: accession: PRJNA182153, control SRA run: SRR1238549, CFIm25 knockdown SRA run: SRR1238551; mouse tissue analysis RNA sequencing: Accession: PRJNA375882, brain SRA runs: SRR5273637 and SRR5273673, liver SRA runs: SRR5273636 and SRR5273672). The BNLx and SHR RNA sequencing that support this study have been deposited in NCBI SRA with the primary accession code GSE166117 (BNLx: GSM5061950-52; SHR: GSM5061947-49).

## Code availability

The software aptardi[98] is maintained on its GitHub repository (https://github.com/luskry/aptardi).

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

## Acknowledgements

We thank Drs. Richard A. Radcliffe, Paula L. Hoffman, and Peter L. Anderson for their helpful discussions and Spencer Mahaffey for his help managing the data. This research was supported by the following US National Institutes of Health (NIH) grants: NIAAA F31AA027430, NIAAA R24AA013162, and NIDA P30DA044223.

## Author contributions

Conceptualization: R.L and L.S.; methodology: R.L., L.S., B.T., and K.K.; machine learning: L.S., R.L., E.S., and F.B.; data curation: R.L., L.S., and B.T.; implementation: R.L.; analysis: R.L., L.S., B.T., and K.K.; visualization: R.L. and L.S.; software: R.L. and E.S.; funding acquisition: R.L., L.S., and B.T.; supervision: L.S.; writing: R.L. and L.S. with input from all authors

## Competing interests

The authors declare no competing interests.
