## [Peer Review File · Nature Communications]

REVIEWER COMMENTS

Reviewer #1 (Remarks to the Author):

The authors proposed aptardi that leverages both DNA sequence and RNA sequencing in a deep learning machine learning model to predict expressed polyA sites. Compared with an existing tool (TAPAS), the aptardi model has higher sensitivity and accuracy. There are various tools for predicting polyA sites based on RNA-seq or DNA sequence. The main contribution of this study is combining both RNA-seq and DNA sequence. However it has some limitations in practical use. First, it requires model training using multiple inputs of the reference genome, RNA-seq, and known polyA sites. Second, true polyA sites (from matched samples) for model training are not always available or complete. Another limitation is that the resolution of the predicted polyA sites is relatively low (100 bp region).

Major points:

1) The title of this manuscript is a bit exaggerated and inappropriate as the main work of this study is predicting polyA sites of a given sample based on RNA-seq coverage and DNA sequence information. And the deep learning model seems not be the (most) important aspect of this study (it was not mentioned in the abstract). I suggest the authors revise the title to make it more in line with the actual content of the research.

2) The authors only compared their tool (aptardi) with TAPAS, which was reported to be one of the best tools for predicting polyA sites from RNA-seq. However, there are several other latest tools based on deep learning models, such as (Arefeen, et al., 2019; Bogard, et al., 2019), that predict polyA sites from DNA sequence, I would suggest additional comparisons with these tools.

3) The availability and authenticity of true polyA sites (i.e. labels for machine learning) is important for the model training. In this study, the authors used true polyA sites from Derti et al, including UHR and HBR RNA reference samples, as well as brain total RNA. The RNA-seq data of matched samples are from other studies. As we know the polyA site annotation is not complete and may contain false positives. More important, true polyA sites of matched samples are not always available for any RNA-seq sample. Therefore, it is important to evaluate the effect of the model using different polyA site annotations. Currently, there are many polyA site database available, e.g., PolyASite 2.0 and PolyA_DB 3 (Herrmann, et al., 2020; Wang, et al., 2018). I would suggest that the authors use these latest polyA site annotation to further evaluate their model.

4) The deep learning approach here is quite similar to the conventional classification problem using traditional classifiers, both of which are based on various features. The main difference is adding a sequential information by the LSTM neural network. It would be interesting to see whether the deep learning approach would actually enhance the prediction performance compared to traditional classifiers (e.g., SVM, RF etc.).

5) Although the micro-heterogeneity of cleavage sites is a common phenomenon, locating polyA sites is still important in many situations (e.g., accurate quantification, searching for polyA signals). Moreover, for many species, the annotated 3' UTR is already quite short (e.g., Arabidopsis is < 300bp). Consequently, the 100 bp resolution in this study is a bit too low, which may not be suitable for species with shorter 3' UTRs. Since the binning strategy was used, is it possible to increase the resolution by reducing the bin width?

6) TAPAS can predict the position of poly(A) sites. As reported previously, the predicted position of polyA site in TAPAS is quite close to the known polyA sites (see Figure S3 of (Chen, et al., 2019)). The 100 bp region was defined as including a polyA site or not in this study, while TAPAS predicted the position of a polyA site. For fair comparison, does the 3' end of the bin rather than the whole bin from aptardi was used, and the position from TAPAS was used? In Figure 4, there are much higher number of false positives in TAPAS results, I suspect one reason may be the true polyA sites used. In this study, the authors used RNA-seq and true brain polyA sites for model training and test, while TAPAS only used the RNA-seq data. It is possible that the sites predicted from TAPAS were not included in the true brain polyA sites but they may be included in the larger

compendium of known polyA sites. I suggest the authors use the latest polyA site database to examine further the predicted results.

Minor points:

1) It is not clear how the binning was performed along the 3' terminal exon. As far as I understand, first the 3' terminal exon was extended 10,200 bp and the refined (shortened or remained). Then the 3' region was binned into 100 bp bins for feature extraction. In page 27 "a length of 300 bins (30,000 bases total)" -- but 3' terminal exon is not of the same length for all genes, why here is 30,000 bp total? Will the binning extend to the downstream of refined 3' terminal exon if the 3' exon is < 30,000bp? But what if there is another gene located downstream near the 3' terminal exon? I suggest the authors use a schematic diagram to explain the binning process.

2) In page 27, "A total of 778,166 bins were present, of which 42,977 possessed a polyA site." - So the dataset for machine learning is very imbalanced (i.e., much more negative bins (94.5%) than positive bins (5.5%)). Although the authors already used the AP metric, I suggest they add an additional popular metric for imbalanced classification, F-Measure.

3) Figures 2a and 2b are not easy to understand. E.g., what do the row and column of the matrix mean?

4) Figure 3 -- the legend overlaid the grid.

Supplementary Fig. 2 -- the label overlaid the upper left corner of the matrix.

Arefeen, A., Xiao, X. and Jiang, T. (2019) DeepPASTA: deep neural network based polyadenylation site analysis, *Bioinformatics*, 35, 4577-4585.

Bogard, N., et al. (2019) A Deep Neural Network for Predicting and Engineering Alternative Polyadenylation, *Cell*, 178, 91-106.e123.

Chen, M., et al. (2019) A survey on identification and quantification of alternative polyadenylation sites from RNA-seq data, *Briefings in Bioinformatics*, 21, 1261-1276.

Herrmann, C.J., et al. (2020) PolyASite 2.0: a consolidated atlas of polyadenylation sites from 3' end sequencing, *Nucleic Acids Res*, 48, D174-D179.

Wang, R., et al. (2018) PolyA_DB 3 catalogs cleavage and polyadenylation sites identified by deep sequencing in multiple genomes, *Nucleic Acids Res*, 46, D315-d319.

Reviewer #2 (Remarks to the Author):

This paper by Lusk et al. reports a new program named Aptardi, which uses both DNA sequence and RNA-seq reads to predict polyA sites and analyze their regulation (in alternative polyadenylation). Overall, the design is novel and results are promising. There are, however, a few major issues which need to be addressed before the work can be considered for publication.

Major:

- DNA sequence elements are only described in the methods section. This is a very important part of their program. They should be shown in a main figure. More importantly, the relative contributions of these DNA features to polyA site identification should be analyzed and presented.

-The authors used the CFIm25 knockdown data to illustrate the usability of their program. However, this data is known to be an extreme case of alternative polyadenylation. The authors need to show a more modest and 'realistic' situation, for example, difference between different tissues. They should use RNA-seq data and then use the polyA-seq data for validation. Another possibility is to mix the knockdown and control data to various degrees and then apply the method, which may also provide useful statistics on selectivity and sensitivity.

--The program DaPars should also be included for comparison because of its wide use and relevance to the design.

Minor:

- Fig. 1, the graph seems cut off somehow.
- Fig 2. Overall, this figure is quite difficult to grasp. More details should be added to help readers understand the inner working of the program and the relative importance of each feature used.
- Fig. 2A, The authors also used ATTAAA and AGTAAA separately. Their values should be shown as well.

Thank you for the time and effort dedicated to reviewing our manuscript and providing valuable feedback. In response, we have made modifications to the manuscript and are confident that the changes substantially improved the final product. Any changes made to the manuscript were done using track changes. Please see below for a point-by-point response to each comment.

REVIEWER COMMENTS

Reviewer #1 (Remarks to the Author):

The authors proposed aptardi that leverages both DNA sequence and RNA sequencing in a deep learning machine learning model to predict expressed polyA sites. Compared with an existing tool (TAPAS), the aptardi model has higher sensitivity and accuracy. There are various tools for predicting polyA sites based on RNA-seq or DNA sequence. The main contribution of this study is combining both RNA-seq and DNA sequence. However it has some limitations in practical use. First, it requires model training using multiple inputs of the reference genome, RNA-seq, and known polyA sites. Second, true polyA sites (from matched samples) for model training are not always available or complete.

Of the model inputs, we agree that true polyA sites are the most difficult to procure for a specific sample; however, we note that a pre-built model is provided for the user that alleviates the need to train a model on a true polyA sites dataset. In other words, only the DNA sequence and RNA-Seq data are required. Furthermore, the algorithm may theoretically be more accurate with a sample-specific genome, but in all of our performance comparisons we used a reference genome for DNA sequence. The availability of the trained model is noted under the Software section in Methods. While we acknowledge that applying a model built from one dataset on another may result in decreased performance due to polyA site specific phenomenon in a given sample, we showed that the aptardi prediction model built on the Human Brain Reference (HBR) dataset performed comparably across different RNA-Seq libraries, tissues, and species. As more datasets become available, and since the user also has the option of training a new model, future researchers can generate more specific models using the aptardi pipeline if deemed necessary and a true polyA sites dataset is available.

Another limitation is that the resolution of the predicted polyA sites is relatively low (100 bp region).

We acknowledge this shortcoming of aptardi. However, we also note that Chen et al. used up to 150 base cutoffs when determining true positives and noted the difficulty pinpointing precise locations from RNA-Seq, so we believe the 100 base resolution is reasonable. Also see Major Point 5 below for additional comments.

Major points:

1) The title of this manuscript is a bit exaggerated and inappropriate as the main work of this study is predicting polyA sites of a given sample based on RNA-seq coverage and DNA sequence information. And the deep learning model seems not be the (most) important aspect of this study (it was not mentioned in the abstract). I suggest the authors revise the title to make it more in line with the actual content of the research.

We have changed the title to put more emphasis on combining RNA-Seq and DNA sequence and less emphasis on machine learning.

2) The authors only compared their tool (aptardi) with TAPAS, which was reported to be one of the best tools for predicting polyA sites from RNA-seq. However, there are several other latest tools based on deep learning models, such as (Arefeen, et al., 2019; Bogard, et al., 2019), that predict polyA sites from DNA sequence, I would suggest additional comparisons with these tools.

We were able to compare aptardi more easily to APARENT (Bogard et al., 2019). See paragraph 3 under “Improvement of 3’ end annotation in the transcriptome map by aptardi” and Supplementary Table 2.

In contrast, we argue the comparison to DeepPASTA (Arefeen et al., 2019) is not appropriate here. Namely, DeepPASTA makes predictions on a single nucleotide using the surrounding 200 bases. Besides the surrounding sequence, it also requires generating the secondary structure using a different program (RNAShapes). As a result, it is time prohibitive to analyze every single base analyzed by aptardi. We believe one of the major advantages of aptardi is that it globally profiles the transcriptome and can be easily incorporated into current bioinformatics pipelines.

3) The availability and authenticity of true polyA sites (i.e. labels for machine learning) is important for the model training. In this study, the authors used true polyA sites from Derti et al, including UHR and HBR RNA reference samples, as well as brain total RNA. The RNA-seq data of matched samples are from other studies. As we know the polyA site annotation is not complete and may contain false positives. More important, true polyA sites of matched samples are not always available for any RNA-seq sample. Therefore, it is important to evaluate the effect of the model using different polyA site annotations. Currently, there are many polyA site database available, e.g., PolyASite 2.0 and PolyA_DB 3 (Herrmann, et al., 2020; Wang, et al., 2018). I would suggest that the authors use these latest polyA site annotation to further evaluate their model.

While we designed aptardi to make sample-specific predictions and these databases contain expressed polyA sites from multiple samples, we applied aptardi to these datasets for evaluation purposes. We acknowledge Chen et al. did a similar comparison and appreciate the additional insight gained from this evaluation. We added this in the results under the section “Performance of aptardi on other true polyadenylation sites datasets.”

4) The deep learning approach here is quite similar to the conventional classification problem using traditional classifiers, both of which are based on various features. The main difference is adding a sequential information by the LSTM neural network. It would be interesting to see whether the deep learning approach would actually enhance the prediction performance compared to traditional classifiers (e.g., SVM, RF etc.).

We added performance of the SVM and RF on the dataset in the methods section to justify use of the biLSTM.

5) Although the micro-heterogeneity of cleavage sites is a common phenomenon, locating polyA sites is still important in many situations (e.g., accurate quantification, searching for polyA signals). Moreover, for many species, the annotated 3' UTR is already quite short (e.g., Arabidopsis is < 300bp). Consequently, the 100 bp resolution in this study is a bit too low, which may not be suitable for species with shorter 3' UTRs. Since the binning strategy was used, is it possible to increase the resolution by reducing the bin width?

To evaluate the impact of window size on performance, we also built the aptardi prediction model on the HBR dataset using 25, 50, and 150 bp windows in addition to the original 100 bp window and added a section in the methods titled “Choice of bin size.” We note that higher resolution bin sizes may be appropriate depending on the dataset (e.g. RNA-Seq library prep) and added this as a parameter than can be specified.

6) TAPAS can predict the position of poly(A) sites. As reported previously, the predicted position of polyA site in TAPAS is quite close to the known polyA sites (see Figure S3 of (Chen, et al., 2019)). The 100 bp region was defined as including a polyA site or not in this study, while TAPAS predicted the position of a polyA site. For fair comparison, does the 3' end of the bin rather than the whole bin from aptardi was used, and the position from TAPAS was used? In Figure 4, there are much higher number of false positives in TAPAS results, I suspect one reason may be the true polyA sites used. In this study, the authors used RNA-seq and true brain polyA sites for model training and test, while TAPAS only used the RNA-seq data. It is possible that the sites predicted from TAPAS were not included in the true brain polyA sites but they may be included in the larger compendium of known polyA sites. I suggest the authors use the latest polyA site database to examine further the predicted results.

The same 100 base distance cutoff was used to define true positives for TAPAS as depicted in Fig. 4. We added additional text to clarify under “Improvement of 3' end annotation in the transcriptome map by aptardi.”

To assess the impact of base distance cutoff and polyA site annotation data sources, we included analysis for 25, 50, 100, and 150 base distance cutoffs using PolyA site annotations from the HBR PolyA-Seq dataset, PolyASite 2.0 dataset, and PolyA_DB dataset (see Supplementary Table 2). Originally, only the 100 base distance cutoff and the HBR PolyA-Seq datasets were used. Regardless of dataset and cutoff, aptardi maintained a superior positive predictive value compared to TAPAS.

Minor points:

1) It is not clear how the binning was performed along the 3' terminal exon. As far as I understand, first the 3' terminal exon was extended 10,200 bp and the refined (shortened or remained). Then the 3' region was binned into 100 bp bins for feature extraction. In page 27 “a length of 300 bins (30,000 bases total)” -- but 3' terminal exon is not of the same length for all genes, why here is 30,000 bp total? Will the binning extend to the downstream of refined 3' terminal exon if the 3' exon is < 30,000bp? But what if there is another gene located downstream near the 3' terminal exon? I suggest the authors use a schematic diagram to explain the binning process.

We added a schematic diagram to explain the binning process (see Supplementary Fig. 7). The biLSTM model requires all batches (i.e., transcripts) to have the same number of timesteps (i.e., bins). To ensure this, we masked transcripts shorter than 300 bins to 300 bins. These masked bins are essentially invisible with respect to machine learning, i.e., they are not used by the biLSTM when being trained or making predictions. The biLSTM is made aware of masked bins by specifying masking in the model design and using a specific value (here we used 99) to denote masked bins, i.e., all masked bins have 99 for all the features.

2) In page 27, “A total of 778,166 bins were present, of which 42,977 possessed a polyA site.” - So the dataset for machine learning is very imbalanced (i.e., much more negative bins (94.5%) than positive bins (5.5%)). Although the authors already used the AP metric, I suggest they add an additional popular metric for imbalanced classification, F-Measure.

We incorporated F-measures throughout the manuscript.

3) Figures 2a and 2b are not easy to understand. E.g., what do the row and column of the matrix mean?

Per this comment and Reviewer #2 comments on Fig. 2, we changed Fig. 2a and added text to explain Fig. 2b.

4) Figure 3 -- the legend overlaid the grid.
Supplementary Fig. 2 -- the label overlaid the upper left corner of the matrix.

We addressed these issues.

Arefeen, A., Xiao, X. and Jiang, T. (2019) DeepPASTA: deep neural network based polyadenylation site analysis, *Bioinformatics*, 35, 4577-4585.

Bogard, N., et al. (2019) A Deep Neural Network for Predicting and Engineering Alternative Polyadenylation, *Cell*, 178, 91-106.e123.

Chen, M., et al. (2019) A survey on identification and quantification of alternative polyadenylation sites from RNA-seq data, *Briefings in Bioinformatics*, 21, 1261–1276.

Herrmann, C.J., et al. (2020) PolyASite 2.0: a consolidated atlas of polyadenylation sites from 3' end sequencing, *Nucleic Acids Res*, 48, D174-D179.

Wang, R., et al. (2018) PolyA_DB 3 catalogs cleavage and polyadenylation sites identified by deep sequencing in multiple genomes, *Nucleic Acids Res*, 46, D315-d319.

Reviewer #2 (Remarks to the Author):

This paper by Lusk et al. reports a new program named Aptardi, which uses both DNA sequence and RNA-seq reads to predict polyA sites and analyze their regulation (in alternative polyadenylation). Overall, the design is novel and results are promising. There are, however, a few major issues which need to be addressed before the work can be considered for publication.

Major:

- DNA sequence elements are only described in the methods section. This is a very important part of their program. They should be shown in a main figure. More importantly, the relative contributions of these DNA features to polyA site identification should be analyzed and presented.

The relative contributions of the individual DNA sequence features were further explored by generating aptardi models for each DNA sequence feature that either 1) included all other features but the DNA sequence feature or 2) removed all other DNA-derived features but the DNA sequence feature (i.e., all of the RNA-Seq features and the original transcriptome feature were also still included) and evaluating performance on the testing split. These results are presented under “Construction of multi-omics model for identification of polyadenylation sites.”

-The authors used the CFIm25 knockdown data to illustrate the usability of their program. However, this data is known to be an extreme case of alternative polyadenylation. The authors need to show a more modest and ‘realistic’ situation, for example, difference between different tissues. They should use RNA-seq data and then use the polyA-seq data for validation. Another possibility is to mix the knockdown and control data to various degrees and then apply the method, which may also provide useful statistics on selectivity and sensitivity.

We added an analysis comparing mouse brain and liver tissue see “Comparison of aptardi predictions across mouse tissues” in Results and “Mouse tissue analysis” in Methods.

--The program DaPars should also be included for comparison because of its wide use and relevance to the design.

We note that the alternative polyadenylation sites identified by the original study and compared to our algorithm were determined by DaPars. Specifically, the authors experimentally determined expression of a proximal alternative polyadenylation isoform for each of the genes in Fig. 5 and used DaPars to computationally estimate the exact location of these proximal transcripts. The similarity between the original study’s estimates (DaPars) and aptardi’s are listed at the end of the paragraph titled “Aptardi identifies novel sample-specific transcripts missed by current transcriptome reconstruction methods.”

Minor:

- Fig. 1, the graph seems cut off somehow.

We changed the text that was incorrect for the figure.

- Fig 2. Overall, this figure is quite difficult to grasp. More details should be added to help readers understand the inner working of the program and the relative importance of each feature used.

Per this comment and comment below (as well as a comment by Reviewer #1), we changed Fig. 2a and included ATTAAA and AGTAAA and added text for clarification.

- Fig. 2A, The authors also used ATTAAA and AGTAAA separately. Their values should be shown as well.

See above.

REVIEWERS' COMMENTS

Reviewer #1 (Remarks to the Author):

The authors have addressed all my concerns satisfactorily.

Reviewer #2 (Remarks to the Author):

The authors have addressed all my concerns. I have no more issues.

Reviewer #1 (Remarks to the Author):

The authors have addressed all my concerns satisfactorily.

We would like to thank the reviewer for taking the time to read our work and providing valuable insights that improved the final product.

Reviewer #2 (Remarks to the Author):

The authors have addressed all my concerns. I have no more issues.

We would like to thank the reviewer for taking the time to read our work and providing valuable insights that improved the final product.